# Visual Outcome after Intravitreal Anti-VEGF Therapy for Macular Neovascularisation Secondary to Sorsby’s Fundus Dystrophy: A Systematic Review

**DOI:** 10.3390/jcm10112433

**Published:** 2021-05-30

**Authors:** Arthur Baston, Christin Gerhardt, Souska Zandi, Justus G. Garweg

**Affiliations:** 1Swiss Eye Institute, Rotkreuz, and Retina Clinic, Berner Augenklinik am Lindenhofspital, 3012 Bern, Switzerland; arthur.baston@augenklinik-bern.ch (A.B.); christin.gerhardt@augenklinik-bern.ch (C.G.); 2Department of Ophthalmology, Inselspital, Bern University Hospital, University of Bern, 3010 Bern, Switzerland; souskasophie.zandi@insel.ch

**Keywords:** Sorsby’s fundus dystrophy, Sorsby, hereditary retinal dystrophy, choroidal neovascularisation, macular neovascularization, anti-VEGF treatment, long-term FU, treatment outcome

## Abstract

The aim of this paper is to summarise our own and to review published experience regarding the long-term outcome of intravitreal treatment for macular neovascularisation (MNV) secondary to Sorsby’s fundus dystrophy (SFD). A systematic literature search using the MeSH terms [Sorsby] and [anti-vascular endothelial growth factor (VEGF)] was conducted in NCBI/PubMed, Cochrane Central Register of Controlled Trials (CENTRAL), ScienceDirect, Google Scholar and ClinicalTrials.gov to identify publications reporting anti-VEGF treatment outcomes in SFD. Treatment outcomes were extracted for this meta-analysis from 14 publications and an own patient reporting a total of 31 cases with a mean follow-up (FU) of 54 months. Both eyes were affected in ten (32.3%) instances. Heterogenous reporting limited the comparability of the outcomes. All papers in common, however, reported satisfied to excellent responses to anti-VEGF therapy if patients were diagnosed and treated immediately after onset of symptoms. Of 20 eyes, for which visual acuity was reported before and after treatment, five worsened and seven improved by more than 1 line, whereas eight eyes maintained their function by end of the follow up, and 11 eyes (55%) maintained a driving vision (Snellen VA ≥ 0.5). Of six eyes with a VA < 0.5, VA improved in one to VA ≥ 0.5, whereas of 14 eyes with an initial VA ≥ 0.5, this dropped to <0.5 despite therapy. In MNV secondary to SFD, the delay between first symptoms and access to anti-VEGF treatment determines subretinal scar formation and thereby, functional prognosis. If treated early, this is generally favourable under regular controls and a consequent anti-VEGF treatment of MNV activity.

## 1. Introduction

Sorsby‘s fundus dystrophy (SFD) is a rare, autosomal dominant inherited retinal disease with complete penetrance affecting both genders similarly, typically becoming symptomatic after the second decade of life, with an average onset in the 4th to 5th decade of life, leading to severe bilateral vision loss and blindness if left untreated [1,2]. The pathophysiological mechanisms underlying the disease have yet to be identified while it is known to be caused by mutations in the gene encoding tissue inhibitor of metalloproteinases-3 (TIMP3) [3]. TIMP3 regulates remodeling of the extracellular matrix by inhibiting metalloproteases (MMPs) and competes with VEGF in binding to its receptor VEGFR2, thereby inhibiting angiogenesis [4,5,6]. It is expressed by retinal pigment epithelium (RPE) cells and is an element of Bruch‘s membrane in healthy individuals. Altered structure and aggregation of the protein can lead to characteristic accumulations in Bruch’s membrane in SFD patients, resulting in Drusen-like deposits and thickening of the membrane [7,8]. What remains to be discovered is whether the accumulation of TIMP3 directly leads to disruption of Bruch‘s membrane, or indirectly, by the failure to inhibit MMP activity and VEGF-driven angiogenesis, resulting in the development of choroidal neovascularisation (CNV) or, due to the underlying pathophysiology more appropriately synonymously used, macular neovascularisation (MNV) [9,10].

SFD is characterised by the loss of central vision due to the development of a classical MNV (Figure 1a,b), and in the clinical course central geographic atrophy (Figure 2) [11]. Classical MNV was found to be a significant risk factor for a poor long-term prognosis in response to foveal scar formation in aged related macular degeneration [12]. Early symptoms in SFD include metamorphopsia, reduced colour vision, difficulties with dark adaptation and nyctalopia [2,13]. The typical clinical presentation of affected patients also includes drusen, reticular pseudodrusen and peripheral pseudodrusen. The hallmark of the angiogenic switch to macular neovascularisation is subretinal haemorrhage and exudation, whereas disciform macular scarring and central pigment epithelium atrophy represent the late stages (Figure 3) [13,14,15]. Progressive peripheral chorioretinal atrophy (Figure 4) and loss of ambulatory vision may be seen [11,13].

The differential diagnosis in this relatively young population is mostly straight forward with a positive family history and includes other inherited macular dystrophies, presenting an age-related macular degeneration (AMD)-like morphology and secondary MNV pathologies, though these but rarely present bilateral [16]. As there is no causal therapy available, current symptomatic treatment has focused on the management of hemeralopia and neovascular complications. Vitamin A has been used, to some extent, to improve night blindness [3]. While lower doses lack efficacy, high doses increase the risk of hepatotoxicity [17]. The formation of MNV is the main cause of severe visual impairment. Thermal laser photocoagulation of MNV has failed to improve vision, but was found to induced frequent recurrences [11,18]. In the early 2000s, verteporfin became available and photodynamic therapy (PDT) was used to treat subfoveal MNV alone or combined with intravitreal corticosteroids. The effect of PDT on MNV activity was limited and not predictable [11,17,19,20,21,22,23,24]. Five years later, access to intravitreal anti-VEGF therapy provided a new treatment option for different types of MNV, including SFD. Until the advent of anti-VEGF drugs, SFD had a poor prognosis and eventually led to bilateral loss of central vision [11,14]; however, more than a decade later, several reports demonstrated promising long-term results preserving a meaningful VA. This compelled us to review the literature regarding long-term visual outcomes in patients with SFD since the advent of anti-VEGF treatment. We also added the experience of our own patient, who has been treated for the past 18 years in our clinic and retained a VA of 20/20 in his better eye.

## 2. Materials and Methods

A systematic literature search was conducted on 15 February 2021 of the NCBI/PubMed Cochrane Central Register of Controlled Trials (CENTRAL), ScienceDirect, Google Scholar and ClinicalTrials.gov databases using the key and MeSH terms [Sorsby] AND [anti-VEGF OR bevacizumab OR ranibizumab OR aflibercept OR photodynamic] and according to PRISMA (Preferred Reporting Items for Systematic Reviews and Meta-Analyses) guidelines. To ascertain maximal exhaustiveness, cross-checking was performed in the reference lists of all papers, including meta-analyses and systematic reviews, to further identify cases meeting the diagnosis and treatment requirements, but not appearing under the above-mentioned MeSH terms and key words. Only articles and conference abstracts providing sufficient information to allow the assessment of the evolution of visual function with anti-VEGF treatment over a minimal FU of at least 3 months and written in English, German or French were included. 

### 2.1. Eligibility Criteria

Criteria applied for studies to be considered eligible for this meta-analysis were:

Report of single or multiple patient case or cohort study including patients diagnosed with Sorsby’s Fundus Dystrophy published or treated until February 2021;Additional or pre-treatment with corticosteroids or photodynamic therapy was accepted;Reporting of evolution of visual function.

### 2.2. Information Retrieved from the Included Publications

The following parameters were retrieved: authors, publication date, title of the publication, gender of patient(s), age at onset of disease and at treatment initiation, time since diagnosis, family history, treatment history, laterality of affected eyes, evolution of VA under therapy, time gap between symptomatic vision loss and treatment initiation, FU duration after first anti-VEGF injection, total number of injections, additional treatment, and, if provided, genetic mutations. The same was applied to both eyes of our own patient.

Whenever necessary and to contain a uniform data format, we converted VA scores into Snellen decimal VA. For the analysis, data for each affected eye were recorded separately (one line in the table). For maximal completeness of the data sets, data from eyes represented in several citations were composed, if the supplemental articles added additional information on this study.

### 2.3. Assessment of Risk of Bias

Since this systematic review summarises case studies, we decided to integrate raw data instead of effect sizes from those reports with no underlying study design that could be biased. Following, a specific assessment of bias is not applicable. Some selection bias based on the orphan disease diagnosis may indeed be present, since the target population is very narrowly outlined. Our demographic data nevertheless show that we have a range of age in the predicted window (32 to 57 years) as well as a comparably balanced gender ratio (60.9% male). Based thereon, we assume that selection bias might not be a relevant problem.

## 3. Results

The systematic literature search generated a total of 907 records (PRISMA search flow, Figure 5). After exclusion of duplicates and the first screening of titles and abstracts, 21 full-text articles remained. After full-text reading, 14 publications reporting on 30 cases were included in the final analysis. All cases were independently coded by two raters. Interrater reliability was calculated in order to show agreement between the two raters. Cohen’s kappa [25] yielded 92%, indicating a high interrater agreement. Differences in data extraction were resolved by discussion. These data were completed by results of an own case under long-term treatment for SFD.

### Meta-Analysis

The overall FU time was 54 months. Eighteen of the thirty-one patients were case reports (Table 1). These 18 cases (six female, ten male, two unknown) referred to 27 affected eyes. Mean age at onset of MNV was 40 years. Mean VA at onset of MNV was 0.63 and last reported was 0.55 in all 18 cases. Considering only cases with both onset and last reported VA, it sums up to 0.63 and 0.62, respectively. Mean FU time was 52.8 months. VA was reported for all 27 eyes at the end of FU and for 20 of these eyes before and after treatment. Beyond all 27 eyes, 67% maintained a Snellen VA of 0.2 or better, and 51% maintained a value ≥0.5. Beyond the 20 eyes with VA known before and after treatment, five (28%) lost >one line, three (17%) ≥three lines, whereas seven eyes (39%) remained stable (±one line), six (33%) gained >one line, and beyond these, three (17%) gained three or more lines. When comparing patients with immediate (13 eyes) and delayed treatment (five eyes), we found that immediate treatment led to an increase of 0.16 of VA, whereas delayed treatment led to a decrease of 0.38 of VA by the end of observation. It must be considered, however, that VA at onset was better for the delayed treatment group (1.12) compared to the immediate treatment group (0.46).

An additional 13 cases participated in two cohort studies [26,27] (Table 1), of which five were male and three were female (five unknown). The mean age was 45 years, and mean FU was 60 months. Five patients in the first series [26] experienced remarkable protection against severe vision loss over 24 months with anti-VEGF treatment (22.2% of the treated eyes suffered significant vision loss compared to 100% of the eyes in the control group). The second series [27] included nine eyes of eight patients that experienced VA gain with anti-VEGF treatment that was maintained over five years. However, the authors observed a linear decrease in VA of 0.1 logMAR units per year until scar formation.

## 4. Discussion

Secondary MNV is the landmark for the breakdown of VA in SFD. With the introduction of intravitreal anti-VEGF therapy, this previously rapidly blinding disease [11] has, for the first time, found an unprecedented treatment that may preserve useful central vison over many years if initiated early, with 51% of eyes maintaining a reading and driving vision (≥0.5) and 67% of vision allowing reading with reading aids (≥0.2) [11,16,17,19,21,22,28,29,30,31,32,33]. Even under consequent treatment of neovascular activity, the underlying, so far only partially understood heredo-degenerative pathology may progress and result in central geographic atrophy and/or progressive night blindness, for both of which there is currently no treatment available. Fortunately, such progression has not been observed in our patient over the past 18 years (Figure 2, Figure 3 and Figure 4).

Central vision may be maintained as long as a central fibrovascular scar has not developed. In neovascular age related macular degeneration ani-VEGF therapy was found to delay scar formation [12]. A significant number of the published patients (Table 1) retained their central vision at least partially over four to seven years after occurrence of MNV, if anti-VEGF agents were administered shortly after occurrence of MNV. Sanz et al. estimated that the risk of significant visual loss may be reduced by 96% over 24 months, based on their case series of eight eyes if MNV was treated early with anti-VEGF drugs. In their series, 22.2% of treated eyes suffered a significant vision loss, compared to 100% of the eyes in the historical control group [26]. Kaye et al. reported a stabilization of VA with anti-VEGF treatment for MNV in five patients during the five-year observation period. They found, however, a linear annual decrease in VA of 0.1 logMAR units, with macular scar formation as the causative factor [27].

Before the availability of anti-VEGF drugs, treatment aimed at preserving some vision with a series of PDT and parabulbar or intravitreal triamcinolone that may stabilise small lesions as in the right, but not so in the left eye of our patient. The functional success of PDT, however, is unavoidably linked to a significant subretinal fibrovascular scar formation, which in the long term is accompanied by severe vision loss. Fortunately, disease remained quiet in the right eye of our patient over seven years. By then, anti-VEGF treatment had become available. This has allowed to maintain a full vision with a total of 24 intravitreal ranibizumab injections on a PRN basis over meanwhile ten years. Given the long periods of inactive MNV, the treatment burden remained supportable for this patient under a PRN regimen. Long times of inactivity of MNV are not unique, why a treatment following a treat-and-extend strategy in this generally relatively young population cannot generally be recommended.

Though our study is inherently limited by the paucity of retrospectively reported cases, this did not question the tremendous effect of early anti-VEGF therapy. The length of FU period in our and previously published cases proved the long-term efficacy of anti-VEGF treatment for MNV in SFD. Affected patients deserve to be correspondingly educated that there is a good chance to retain useful central vision, and to understand the importance of immediately consulting an ophthalmologist in case of visual irregularities, ideally before severe VA loss is encountered.

Though there exists, in conclusion, no cure for this heredo-degenerative disease, anti-VEGF treatment has dramatically changed the prognosis for patients with Sorsby’s fundus dystrophy. The visual function may be preserved in the vast majority for a significant period of the patients’ lives. More than half of the patients will maintain a driving and reading vision if macular neovascularisation is diagnosed and treated early.

## Figures and Tables

**Figure 1 jcm-10-02433-f001:**
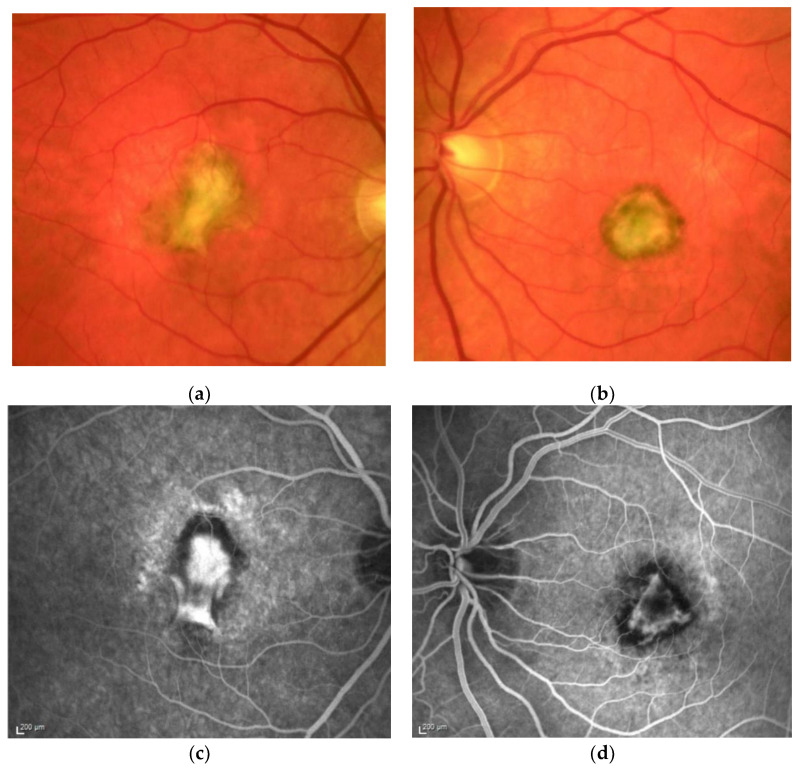
M, 35 years, M. Sorsby. Clinical image of both eyes with a significant submacular fibrovascular lesion after three courses of photodynamic therapy in the right (**a**) and four in the left eye (**b**), prior to the start of intravitreal therapy. (second panel). Same patient, fluorescein angiography (R middle (**c**), L early arteriovenous phase (**d**)) confirming a low-active predominantly classic macular neovascularisation.

**Figure 2 jcm-10-02433-f002:**
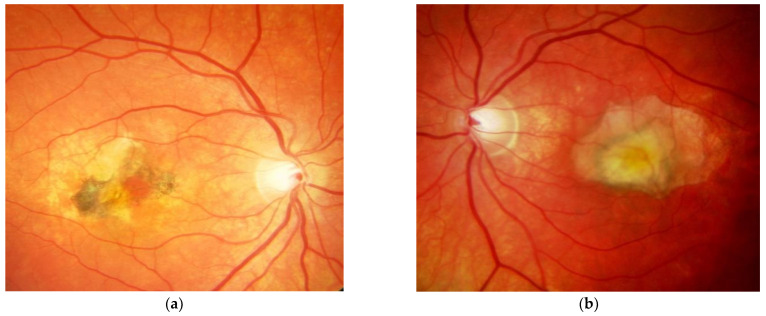
Same patient, 5 years later. First reactivation of macular neovascularisation evidenced by vision loss and a small macular hemorrhage as well as newly present intraretinal fluid in OCT in the right eye (**a**) and macular pigment atrophy in both eyes (**a**,**b**).

**Figure 3 jcm-10-02433-f003:**
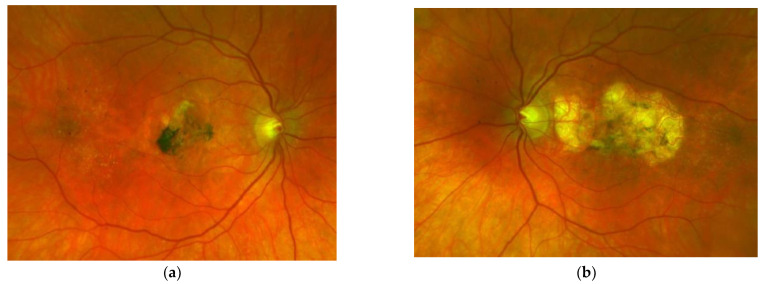
Same patient, 2016, 10 years after the start of intravitreal therapy; no lesion activity after 22 intravitreal Ranibizumab injections in the right eye (**a**) and a remarkable progressive macular atrophy despite a stable scar in his left eye (**b**).

**Figure 4 jcm-10-02433-f004:**
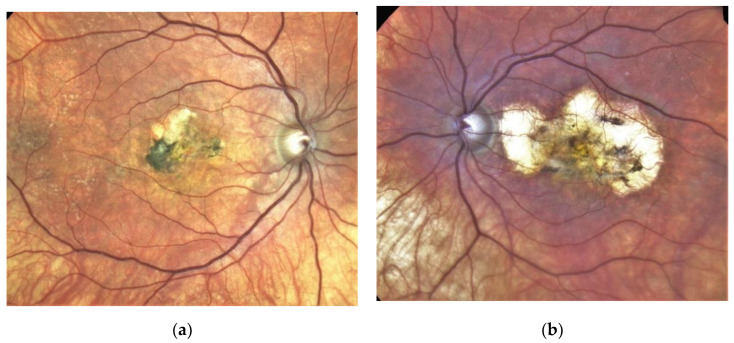
Same patient, 2021, meanwhile 51 years old. Eighteen years after diagnosis and 15 years after the start of intravitreal therapy visual acuity was maintained at Snellen 1.0 (20/20) in his right (**a**) and 0.16 (32/200) in his left eye (**b**), though reading and contrast-enhancing optical aids are required for near visual performance; no lesion activity after 22 intravitreal Ranibizumab injections in the right eye and a widely unchanged macular situation. Progressive macular scarring in both eyes. Upper panel: Clinical pictures of R + L eye (**a**,**b**), second panel, redfree picture and OCT of the right eye (**c**), bottom same, left eye (**d**). The arrows in redfree frames on the left side in Figure 4c,d indicate the location of the line scans on the right side. Note the progression of severity and extension of RPE changes, Drusen formation and choroidal sclerosis during the observation period. With consequent clinical controls and Ranibizumab treatment immediately upon first signs of lesion reactivation, his quality of life is perceived as excellent, he can follow his daily professional and private activities without relevant restrictions.

**Figure 5 jcm-10-02433-f005:**
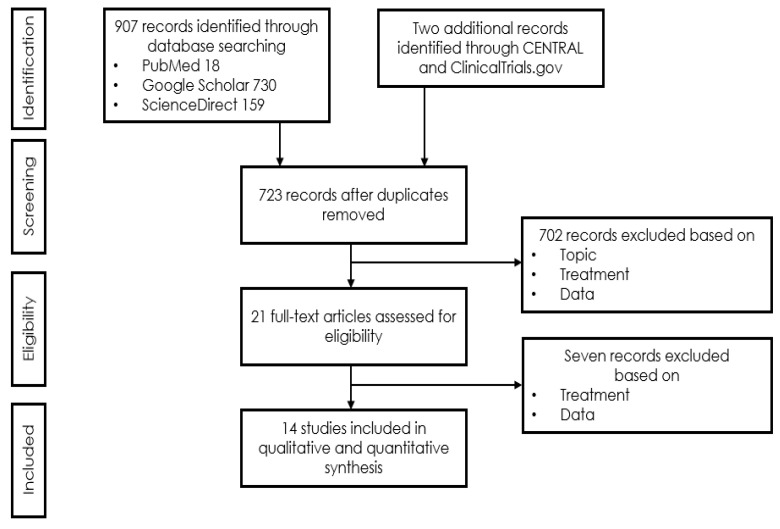
PRISMA (Preferred Reporting Items for Systematic Reviews and Meta-Analyses) search flow.

**Table 1 jcm-10-02433-t001:** Treatment outcomes of case and cohort studies with Sorsby’s fundus dystrophy, part I.

Patient	First Author	Year of Publication		Gender	NR of Eyes	Age at Onset		Family History Positive		Prior Treatment		VA before Onset	Eye
1	Sivaprasad	2008		m	1	nr		yes		PDT		nr	nr
2	Gemenetzi	2011		f	2	34		yes		no		nr	r
2	Gemenetzi	2011		f		37		yes		PDT		1.00	l
3	Gemenetzi	2011		f	1	44		yes		no		1.00	r
4	Gray	2012		f	1	38		yes		no		nr	l
5	Balaskas	2013		m	1	41		nr		no		1.25	r
6	Copete-Piqueras	2013		m	2	32		nr		no		nr	r
6	Copete-Piqueras	2013		m		32		nr		no		nr	l
7	Fung	2013		m	1	44		yes		no		1.00	r
8	Kapoor	2013		m	2	57		yes		no		1.25	r
8	Kapoor	2013		m		57		yes		no		1.00	l
9	Gliem	2015		nr	1	54		yes		no		1.00	l
10	Gliem	2015		nr	1	56		yes		no		1.00	r
11	Gliem	2015		m	1	45		yes		no		1.00	r
12	Keller	2015		m	2	32		yes		nr		nr	r
12	Keller	2015		m		32		yes		PDT		nr	l
13	Keller	2015		m	2	28		yes		no		nr	r
13	Keller	2015		m		28		yes		no		nr	l
14	Mohla	2016		f	1	52		nr		no		0.63	r
15	Menassa	2017		m	2	44		yes		no		1.60	r
15	Menassa	2017		m		38		yes		no		nr	l
16	Tsokolas	2020		f	2	34		yes		no		nr	r
16	Tsokolas	2020		f		37		yes		PDT		nr	l
17	Tsokolas	2020		f	2	36		yes		no		1.25	r
17	Tsokolas	2020		f		38		yes		no		1.00	l
18	Own patient	2004		m	2	33		yes		no		nr	r
18	Own patient	2004		m		33		yes		no		nr	l
19–23 *	Kaye	2017		nr	5	nr		nr		nr		nr	nr
24–31 *	Sanz	2013		62.5% m	9	45.3 (6.9)		nr		nr		nr	nr
Mean				62.5% m	41	41.2		48.4%		9.8%		1.08	
**Patient**	**First Author**	**VA at Onset**	**Treatment Delay (Months)**	**Last VA**	**Follow-Up after Onset of Anti-VEGF Treatment (Months)**		**Total Number of Intravitreal Injections**		**Drug**		**Mutation**		
1	Sivaprasad	0.50	2	0.50	6		2		2 Bev		Ser181Cys		
2	Gemenetzi	0.10	0	0.16	33		6		6 Bev		p.S204C		
2	Gemenetzi	1.60	0.75	1.25	5		3		3 Bev		p.S204C		
3	Gemenetzi	0.10	0	1.00	3		1		1 Bev		p.S204C		
4	Gray	1.00	1.5	1.00	13		3		3 Bev		Ser181Cys		
5	Balaskas	0.16	nr	0.40	27		14		14 Ran		c.610A4T (p.Ser204Cys)		
6	Copete-Piqueras	0.63	0	1.00	6		1		1 Ran		mutations in Exon 5 of gene 22.12.3		
6	Copete-Piqueras	0.80	0	1.00	6		1		1 Ran		mutations in Exon 5 of gene 22.12.3		
7	Fung	0.63	0	0.80	48		6		6 Bev, PDT		Tyr159Cys		
8	Kapoor	0.50	0	0.10	55		8		8 Bev, several Bev-Dex		normal coding sequence (codons 124–188 of the mature protein)		
8	Kapoor	0.63	0	0.40	77		31		8 Bev, min. 18 Bev-Dex, 5 Ran, PDT		normal coding sequence (codons 124–188 of the mature protein)		
9	Gliem	0.80	0	1.00	12		1		1 Bev		c.530A > G (p.Tyr200Cys)		
10	Gliem	0.63	0	1.00	8		nr		multiple Bev		c.530A > G (p.Tyr200Cys)		
11	Gliem	nr	0	1.00	nr		35		35 Bev		c.545A > G(p.Tyr182Cys)		
12	Keller	nr	nr	0.70	60		nr		several Ran and Bev		nr		
12	Keller	nr	nr	0.03	60		3		PDT, 3 Ran		nr		
13	Keller	nr	nr	0.10	48		nr		Multiple Ran		nr		
13	Keller	1.00	nr	0.20	48		nr		Multiple Ran		nr		
14	Mohla	0.10	0	0.32	7		2		2 Bev		p.Arg204Cys		
15	Menassa	1.25	0.3	0.80	6		5		5 Ran		c.610A > T		
15	Menassa	nr	nr	0.10	nr		6		6 Ran		c.610A > T		
16	Tsokolas	0.10	0	0.08	144		5		5 Bev		Ser204Cys		
16	Tsokolas	1.25	1	0.16	108		79		79 Bev		Ser204Cys		
17	Tsokolas	nr	4	0.06	72		24		24 Bev		Ser204Cys		
17	Tsokolas	nr	0	0.50	60		42		42 Bev		Ser204Cys		
18	Own patient		0	1.0	192		24		3 PDT, Tri, 24 Ran		mutation in the TIMP3 gene		
18	Own patient		0	0.16	192		9		4 PDT, multiple Tri, 9 Ran		mutation in the TIMP3 gene		
19–23 *	Kaye	0.8 (0.8)	nr	0.2 (0.4)	Min. 60		16		Bev		mutation in tissue inhibitor of metalloproteinases-3 (TIMP3)		
24–31 *	Sanz	0.25 (0.2)	nr	nr	nr		9.11 (6.01)		Bev, Ran		p.Ser204Cys		
Mean		0.56	0.45	0.49	54		12.78						

Abbreviations: nr, not reported; VA, Snellen visual acuity; FU, follow-up; f, female; m, male; r, right eye; l, left eye; nr, not reported; Bev, bevacizumab; Dex, dexamethasone; Ran, ranibizumab; PDT, photodynamic therapy; Tri, triamcinolone. * Cohort studies: values are reported as mean (standard deviation).

## Data Availability

No new data were created or analyzed in this study. Data sharing is not applicable to this article.

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
