# Peer review of "Visual Outcome after Intravitreal Anti-VEGF Therapy for Macular Neovascularisation Secondary to Sorsby’s Fundus Dystrophy: A Systematic Review"

_jcm, 2021, doi:10.3390/jcm10112433_

Round 1
Reviewer 1 Report
I am very glad the authors wrote this paper. The results from this study represent a significant and a large step forward in understanding the mechanisms pathogenesis, diagnosis and treatment of macular neovascularisation secondary to Sorsby’s fundus dystrophy.
However, several residual issues must still be addressed before the manuscript can be considered for publication by the Editor.
The introduction would benefit from hypotheses providing a rationale.
In the methods, a number of unclear circumstances can be found which have to be carefully clarified.
The review protocol is available in PROSPERO?.
what are the Eligibility criteria?. They are not specified in the manuscript.
Risk-of-bias assessment, how it is performed?
Results
The tables present the data nicely but a bit more explanation of the content and implications of the data in the tables would be beneficial.
Discussion
I would like to be able to distinguish a first paragraph with a summary of what was sought and what was found in this work, a discussion of the relevant findings together with what was found in other works, an explanation of the meaning of these findings, an explanation of the implications for practice clinic and suggestions for future research.
Conclusions
Should provide a conclusion
Author Response
JCM-1213515
Response to the reviewers:
We’d like to thank the authors for their generably favourable votes. Below, you will find a point-by-point response to the single comments:
Reviewer: The introduction would benefit from hypotheses providing a rationale.
Response: We fully agree with the reviewer. Therefore, we placed such at the end of the introduction, now reading: “Until the advent of anti-VEGF drugs, SFD had a poor prognosis and eventually led to bilateral loss of central vision [11, 14]; however, more than a decade later, several reports demonstrated promising long-term results preserving a meaningful VA. This compelled us to review the literature regarding long-term visual outcomes in patients since the advent of SFD with anti-VEGF treatment.”
Reviewer: In the methods, a number of unclear circumstances can be found which have to be carefully clarified. The review protocol is available in PROSPERO?
Response: we did not register the meta-analysis in PROSPERO, but thank the reviewer for asking. We tried to register retrospectively, which was not possible.
Reviewer: what are the Eligibility criteria?. They are not specified in the manuscript.
Response: Thanks for pointing on this weakness. Eligibility criteria were now added, the text reading: “2.1. Eligibility criteria:
Criteria applied for studies to be considered eligible for this meta-analysis were:
Report of single or multiple patients case or cohort study including patients diagnosed with Sorsby’s Fundus Dystrophy published or treated until February 2021;
Additional or pre-treatment with corticosteroids or photodynamic therapy was also accepted;
Reporting of evolution of visual function.”
Reviewer: The tables present the data nicely but a bit more explanation of the content and implications of the data in the tables would be beneficial.
Response: While the respective most important data are summarized in the Results chapter, meta-analysis section, we attempted to exclude repeating the data from the table in the text. The most important findings of this meta-analysis consequently are displayed at the most prominent place in the discussion, in the very first paragraph, now reading: “Secondary MNV is the landmark for the breakdown of VA in SFD. With the introduction of intravitreal anti-VEGF therapy, this previously rapidly blinding disease [11] has, for the first time, found an unprecedented treatment that may preserve useful central vison over many years if initiated early, with 51% of eyes maintaining a reading and driving vision (≥ 0.5) and 67% of vision allowing reading with reading aids (≥ 0.2) [11, 16, 17, 19, 21, 22, 28-33].”
Reviewer: I would like to be able to distinguish a first paragraph with a summary of what was sought and what was found in this work, a discussion of the relevant findings together with what was found in other works, an explanation of the meaning of these findings, an explanation of the implications for practice clinic and suggestions for future research.
Response: We fully agree that much could be hypothesized based on this meta-analysis of published data. Three points are evident:
1. Anti-VEGF treatment has dramatically changed the prognosis for patients with Sorsby’s disease.
2. Only an early diagnosis and treatment as soon as possible after conversion to neovascular disease provide the chance of maintaining reading vision for a significant period of the patient life.
3. Even under consequent treatment of the MNV, the underlying degenerative pathology will progress, resulting in a. night blindness and b. central geographic atrophy, for both of which there is currently no treatment available. We tried to reflect this in the re-worded 1st paragraph of the discussion, now reading: “With the introduction of intravitreal anti-VEGF therapy, this previously rapidly blinding disease [11] has, for the first time, found an unprecedented treatment that may preserve useful central vison over many years if initiated early, with 51% of eyes maintaining a reading and driving vision (≥ 0.5) and 67% of vision allowing reading with reading aids (≥ 0.2) [11, 16, 17, 19, 21, 22, 28-33]. Even under consequent treatment of neovascular activity, the underlying, so far only partially understood heredo-degenerative pathology may progress and result in central geographic atrophy and/or progressive night blindness, for both of which there is currently no treatment available. Fortunately, such progression has not been observed in our patient over the past 18 years (Figures 2-4).”
Reviewer: Should provide a conclusion
Response: We have added a conclusion, reading: “Though there exists, in conclusion, no cure for this heredo-degenerative disease, anti-VEGF treatment has dramatically changed the prognosis for patients with Sorsby’s fundus dystrophy. The visual function may be preserved in the vast majority for a significant period of the patients’ lives. More than half of the patients will maintain a driving and reading vision for many years if macular neovascularization is diagnosed treated early.”
Reviewer 2:
Reviewer: This is an interesting systematic review of studies that have assessed the outcome of anti-VEGF treatment in patients with Sorsbys Fundus dystrophy. The primary limitation of this review is the small number of available and appropriate cases with adequate reporting, in part a consequence of the rarity of Sorsbys disease. This paper will be of particular interest to clinicians to evaluate the benefit of anti-VEGF administration and the importance in limiting treatment delay.
Response: Thanks for this favourable statement.
Reviewer: Line 30-31. Wording of this last sentence could be improved to ensure meaning is clear
Response: We hope that the meaning became more concise with re-wording to: “If treated early, this is generally favourable under regular controls and a consequent anti-VEGF treatment of MNV activity.”
Reviewer: Line 38: Whilst it is true that symptoms have not be reported prior to the second decade, the average onset is the 4th to 5th decade of life, as supported by your own report. It would be more appropriate and informative to give the average age of onset here as very few patients have experienced symptoms before 30 years old.
Response: Since the vast majority of cases become symptomatic after age of 30 yeras, as stated, we changed the sentence to “Sorsby`s fundus dystrophy (SFD) is a rare, autosomal dominant inherited retinal disease with complete penetrance affecting both genders similarly, typically becoming symptomatic after the second decade of life, with an average onset in the 4th to 5th decade of life, leading to severe bilateral vision loss and blindness if left untreated [1, 2].” though we are sorry not to be able of including the anonymous reviewer’s own work.
Reviewer: Line 39-41: I would restructure this sentence to state that the disease is caused by TIMP3 first, followed by explaining that the pathophysiological mechanisms have yet to be identified.
Response: Though we felt that the original text well met the reviewer postulate, we changed the sentence to “The pathophysiological mechanisms underlying the disease have yet to be identified while it is known to be caused by mutations in the gene encoding tissue inhibitor of metalloproteinases-3 (TIMP3) [[3].”
Reviewer: Line 44: grammar: I think 'is' is missing prior to 'an element of BrM'. Without this insertion, it reads that TIMP3 is produced by RPE and the Bruch's membrane.
Response: Thanks for pointing on this, now changed accordingly
Reviewer: Line 45: To my knowledge, it is not currently known whether any altered function of TIMP3 (i.e. MMP/VEGF/ADAM inhibition) as a result of mutations directly contributes to accumulation of TIMP3 in BrM. It is more likely that altered structure/accumulation of TIMP3 contributes to changes in function. It would be more suitable here to say 'altered structure and aggregation'.
Response: Correct, changed as requested.
Reviewer: Line 55: 'Adaptation' rather than 'Adaption'
Response: thanks, corrected
Reviewer: Line 98: spelling: 'Induce' rather than 'induced'
Response: thanks, corrected
Reviewer: Line 99: 1st mention of PDT so need (photodynamic therapy) after abbreviation
Response: thanks, added as requested
Reviewer: Table 1. Part II: Patient 6, mutation: 'Exon 5' not 'axon 5'
Response: Thanks for identifying, corrected
Reviewer: Line 191: 'You have the subheading here: 3.2. evolution of macular neovascularisation and retinal degeneration over time' but there is no text for this section. In addition to VA, it would be useful to have more information on the effect of treatment on the outcomes mentioned in the title.
Response: Thanks for this critical comment. We formerly had a corresponding chapter which the editorial staff asked us to remove. Erroneously maintained headline now deleted. Regarding treatment outcomes, we would indeed love to be more precise, but even visual acuity was not systematically reported prior to and after treatment. Currently, there is no more information available regarding treatment effects on outcomes than what has been displayed in table and Results text.
Reviewer: Line 219-220: English: consider re-wording
Response: With an attempt to improve the meaning, the sentence was re-worded to “By then, anti-VEGF treatment had become available. This has allowed to maintain a full vision with a total of 24 intravitreal ranibizumab injections on a PRN basis over more than ten years. Given the long periods of inactive MNV, the treatment burden remained supportable for this patient under a PRN regimen.”
Reviewer: Line 220: remove 'meanwhile'
Response: Replaced by “more than”
Reviewer: Line 222: Replace 'Times' with 'periods'
Response: done
Reviewer: Line 222-223: English: consider re-wording
Response: Re-worded to “Long periods of inactivity of MNV are not unique. A treatment consequently following a treat-and-extend strategy in this generally relatively young population thus cannot generally be recommended.”
Reviewer: In the abstract (Lines 28-29) you state that the delay in anti-VEGF treatment determines subretinal scar formation. Could this be mentioned in the introduction with appropriate references or discussed further in the discussion (again with suitable references). You mention reports of scarring in patients from the Kaye et al., study. Did any of the other studies look at scarring development or progression following anti-VEGF treatment?
Response: mentioned now in the introduction: “Classical MNV was found to be a significant risk factor for a poor long-term prognosis in response to foveal scar formation in aged related macular degeneration [12]” References for this statement, however, for this tenet are limited to neovascular AMD [Daniel E, Toth CA, et al. Risk of scar in the comparison of age-related macular degeneration treatments trials. Ophthalmology. 2014 Mar;121(3):656-66. doi: 10.1016/j.ophtha.2013.10.019]. Therefore, we preferred not to discuss this further. As mentioned, the paper by Kaye and Lotery has been the only reporting long-term outcomes and subretinal scar formation in SFD.
Reviewer: Is it possible to assess any differences in VA outcomes in patients depending on the specific drug administered, age/sex of patient?
Response: We carefully considered looking at potential outcome factors, but the number of published cases does not provide a sufficient power to draw any supportable conclusion. This holds namely true for a comparison of drugs and treatment protocols, but also does not allow a conclusion regarding age or gender.
Reviewer 2 Report
This is an interesting systematic review of studies that have assessed the outcome of anti-VEGF treatment in patients with Sorsbys Fundus dystrophy. The primary limitation of this review is the small number of available and appropriate cases with adequate reporting, in part a consequence of the rarity of Sorsbys disease. This paper will be of particular interest to clinicians to evaluate the benefit of anti-VEGF administration and the importance in limiting treatment delay.
Abstract:
Line 30-31. Wording of this last sentence could be improved to ensure meaning is clear
Introduction:
Line 38: Whilst it is true that symptoms have not be reported prior to the second decade, the average onset is the 4th to 5th decade of life, as supported by your own report. It would be more appropriate and informative to give the average age of onset here as very few patients have experienced symptoms before 30 years old.
Line 39-41: I would restructure this sentence to state that the disease is caused by TIMP3 first, followed by explaining that the pathophysiological mechanisms have yet to be identified.
Line 44: grammar: I think 'is' is missing prior to 'an element of BrM'. Without this insertion, it reads that TIMP3 is produced by RPE and the Bruch's membrane.
Line 45: To my knowledge, it is not currently known whether any altered function of TIMP3 (i.e. MMP/VEGF/ADAM inhibition) as a result of mutations directly contributes to accumulation of TIMP3 in BrM. It is more likely that altered structure/accumulation of TIMP3 contributes to changes in function. It would be more suitable here to say 'altered structure and aggregation'.
Line 55: 'Adaptation' rather than 'Adaption'
Line 98: spelling: 'Induce' rather than 'induced'
Line 99: 1st mention of PDT so need (photodynamic therapy) after abbreviation
Table 1. Part II: Patient 6, mutation: 'Exon 5' not 'axon 5'
Line 191: 'You have the subheading here: 3.2. evolution of macular neovascularisation and retinal degeneration over time' but there is no text for this section. In addition to VA, it would be useful to have more information on the effect of treatment on the outcomes mentioned in the title.
Line 219-220: English: consider re-wording
Line 220: remove 'meanwhile'
Line 222: Replace 'Times' with 'periods'
Line 222-223: English: consider re-wording
Points of discussion:
- In the abstract (Lines 28-29) you state that the delay in anti-VEGF treatment determines subretinal scar formation. Could this be mentioned in the introduction with appropriate references or discussed further in the discussion (again with suitable references). You mention reports of scarring in patients from the Kaye et al., study. Did any of the other studies look at scarring development or progression following anti-VEGF treatment?
- Is it possible to assess any differences in VA outcomes in patients depending on the specific drug administered, age/sex of patient?
Author Response
JCM-1213515
Response to reviewer 2:
Reviewer: This is an interesting systematic review of studies that have assessed the outcome of anti-VEGF treatment in patients with Sorsbys Fundus dystrophy. The primary limitation of this review is the small number of available and appropriate cases with adequate reporting, in part a consequence of the rarity of Sorsbys disease. This paper will be of particular interest to clinicians to evaluate the benefit of anti-VEGF administration and the importance in limiting treatment delay.
Response: Thanks for this favourable statement.
Reviewer: Line 30-31. Wording of this last sentence could be improved to ensure meaning is clear
Response: We hope that the meaning became more concise with re-wording to: “If treated early, this is generally favourable under regular controls and a consequent anti-VEGF treatment of MNV activity.”
Reviewer: Line 38: Whilst it is true that symptoms have not be reported prior to the second decade, the average onset is the 4th to 5th decade of life, as supported by your own report. It would be more appropriate and informative to give the average age of onset here as very few patients have experienced symptoms before 30 years old.
Response: Since the vast majority of cases become symptomatic after age of 30 yeras, as stated, we changed the sentence to “Sorsby`s fundus dystrophy (SFD) is a rare, autosomal dominant inherited retinal disease with complete penetrance affecting both genders similarly, typically becoming symptomatic after the second decade of life, with an average onset in the 4th to 5th decade of life, leading to severe bilateral vision loss and blindness if left untreated [1, 2].” though we are sorry not to be able of including the anonymous reviewer’s own work.
Reviewer: Line 39-41: I would restructure this sentence to state that the disease is caused by TIMP3 first, followed by explaining that the pathophysiological mechanisms have yet to be identified.
Response: Though we felt that the original text well met the reviewer postulate, we changed the sentence to “The pathophysiological mechanisms underlying the disease have yet to be identified while it is known to be caused by mutations in the gene encoding tissue inhibitor of metalloproteinases-3 (TIMP3) [[3].”
Reviewer: Line 44: grammar: I think 'is' is missing prior to 'an element of BrM'. Without this insertion, it reads that TIMP3 is produced by RPE and the Bruch's membrane.
Response: Thanks for pointing on this, now changed accordingly
Reviewer: Line 45: To my knowledge, it is not currently known whether any altered function of TIMP3 (i.e. MMP/VEGF/ADAM inhibition) as a result of mutations directly contributes to accumulation of TIMP3 in BrM. It is more likely that altered structure/accumulation of TIMP3 contributes to changes in function. It would be more suitable here to say 'altered structure and aggregation'.
Response: Correct, changed as requested.
Reviewer: Line 55: 'Adaptation' rather than 'Adaption'
Response: thanks, corrected
Reviewer: Line 98: spelling: 'Induce' rather than 'induced'
Response: thanks, corrected
Reviewer: Line 99: 1st mention of PDT so need (photodynamic therapy) after abbreviation
Response: thanks, added as requested
Reviewer: Table 1. Part II: Patient 6, mutation: 'Exon 5' not 'axon 5'
Response: Thanks for identifying, corrected
Reviewer: Line 191: 'You have the subheading here: 3.2. evolution of macular neovascularisation and retinal degeneration over time' but there is no text for this section. In addition to VA, it would be useful to have more information on the effect of treatment on the outcomes mentioned in the title.
Response: Thanks for this critical comment. We formerly had a corresponding chapter which the editorial staff asked us to remove. Erroneously maintained headline now deleted. Regarding treatment outcomes, we would indeed love to be more precise, but even visual acuity was not systematically reported prior to and after treatment. Currently, there is no more information available regarding treatment effects on outcomes than what has been displayed in table and Results text.
Reviewer: Line 219-220: English: consider re-wording
Response: With an attempt to improve the meaning, the sentence was re-worded to “By then, anti-VEGF treatment had become available. This has allowed to maintain a full vision with a total of 24 intravitreal ranibizumab injections on a PRN basis over more than ten years. Given the long periods of inactive MNV, the treatment burden remained supportable for this patient under a PRN regimen.”
Reviewer: Line 220: remove 'meanwhile'
Response: Replaced by “more than”
Reviewer: Line 222: Replace 'Times' with 'periods'
Response: done
Reviewer: Line 222-223: English: consider re-wording
Response: Re-worded to “Long periods of inactivity of MNV are not unique. A treatment consequently following a treat-and-extend strategy in this generally relatively young population thus cannot generally be recommended.”
Reviewer: In the abstract (Lines 28-29) you state that the delay in anti-VEGF treatment determines subretinal scar formation. Could this be mentioned in the introduction with appropriate references or discussed further in the discussion (again with suitable references). You mention reports of scarring in patients from the Kaye et al., study. Did any of the other studies look at scarring development or progression following anti-VEGF treatment?
Response: mentioned now in the introduction: “Classical MNV was found to be a significant risk factor for a poor long-term prognosis in response to foveal scar formation in aged related macular degeneration [12]” References for this statement, however, for this tenet are limited to neovascular AMD [Daniel E, Toth CA, et al. Risk of scar in the comparison of age-related macular degeneration treatments trials. Ophthalmology. 2014 Mar;121(3):656-66. doi: 10.1016/j.ophtha.2013.10.019]. Therefore, we preferred not to discuss this further. As mentioned, the paper by Kaye and Lotery has been the only reporting long-term outcomes and subretinal scar formation in SFD.
Reviewer: Is it possible to assess any differences in VA outcomes in patients depending on the specific drug administered, age/sex of patient?
Response: We carefully considered looking at potential outcome factors, but the number of published cases does not provide a sufficient power to draw any supportable conclusion. This holds namely true for a comparison of drugs and treatment protocols, but also does not allow a conclusion regarding age or gender.
This manuscript is a resubmission of an earlier submission. The following is a list of the peer review reports and author responses from that submission.